# Group additivity-Pourbaix diagrams advocate thermodynamically stable nanoscale clusters in aqueous environments

Lindsay A. Wills[1], Xiaohui Qu[2], I-Ya Chang[1], Thomas J.L. Mustard[1], Douglas A. Keszler[1], Kristin A. Persson[2] & Paul Ha-Yeon Cheong[1]

The characterization of water-based corrosion, geochemical, environmental and catalytic processes rely on the accurate depiction of stable phases in a water environment. The process is aided by Pourbaix diagrams, which map the equilibrium solid and solution phases under varying conditions of pH and electrochemical potential. Recently, metastable or possibly stable nanometric aqueous clusters have been proposed as intermediate species in non-classical nucleation processes. Herein, we describe a Group Additivity approach to obtain Pourbaix diagrams with full consideration of multimeric cluster speciation from computations. Comparisons with existing titration results from experiments yield excellent agreement. Applying this Group Additivity-Pourbaix approach to Group 13 elements, we arrive at a quantitative evaluation of cluster stability, as a function of pH and concentration, and present compelling support for not only metastable but also thermodynamically stable multimeric clusters in aqueous solutions.

[1] Department of Chemistry, Oregon State University, 153 Gilbert Hall, Corvallis, Oregon 97331-4003, USA. [2] Department of Materials Science and Engineering, UC Berkeley, Berkeley, California 94720, USA. Correspondence and requests for materials should be addressed to D.A.K. (email: douglas.keszler@oregonstate.edu) or to K.A.P. (email: kristinpersson@berkeley.edu) or to P.H.-Y.C. (email: paulc@science.oregonstate.edu).

Cluster formation in aqueous solutions have attracted renewed interest during the last decade[1]. Aqueous inorganic clusters are nanoscale molecular solvates composed of several metal ions linked via oxo and hydroxo ligands. They are crucial in a diverse range of chemical processes such as solution deposition of device thin films[2], biomineralization[3,4] and the natural cycling of minerals through soil, water and biomass[5]. Aqueous clusters are also known to be causative agents of neurodegenerative disease and image enhancement additives[6]. Recently, stable or metastable aqueous cluster formation have been suggested to play an important role in an alternative nucleation process as compared to classical nucleation theory (CNT)[3,7–9]. In this scenario, nucleation towards a solid bulk state proceeds via aggregation of solvated clusters, and not via a one-at-a-time attachment of monomeric species to a critical nucleus as assumed in CNT[10]. The occurrence of nanoscale aqueous clusters, in systems such as $CaCO_3$ (refs 3,7,11,12), $AlPO_4$ (ref. 13), Ga (ref. 14), Fe (ref. 15) Fe oxyhydroxides (ref. 16), Hf/Zr (ref. 17) and Al (ref. 18) has been inferred from a variety of experimental visualization, speciation and spectroscopic observations. From the evidence, it is clear that these clusters (i) form through a reversible process[3,19] whose speciation is governed by equilibria of molecular species in solution[3] and (ii) are stable in solution for an extended time[3,7,8]. While the majority of published Pourbaix diagrams[20–22] exhibit exclusively monomeric aqueous ion stability domains, growing thermochemical data on clusters is being incorporated into speciation and Pourbaix analyses. As examples, we note that stable multimeric species, beyond dimers and trimers, are suggested for Mo, W and V[22] and for Bi[20] at 0.1 M and high potential. On the other hand, known cluster-forming systems such as Al, Fe, Ga, Hf and so on, exhibit monomeric aqueous ion stability in these reference works. Hence, there are still many questions surrounding nanometric cluster formation in solution, for example, as to the structure, the stability, the concentration ranges, counter-ion effects and solvation structure as well as the role of clusters in the nucleation process[23]. Theory can provide a complementary means to elucidate some of these questions. For example, quantum mechanical (QM) studies of the structure, electronic and spectroscopic properties, absorption and reaction sites for clusters are well established[24,25]. Recently, molecular dynamics simulations have been used to unravel solute-solvent and ion-pairing interactions as well as the dynamics of specific clusters[24,26,27].

In this work, we present two distinct theoretical procedures, which together provide an efficient and theoretically rigorous route towards obtaining Pourbaix diagrams with full consideration of cluster stability and speciation from first-principles. First, we reveal a group additivity (GA) method that can predict the Gibbs free energies of aqueous oxo/hydroxo metal clusters in excellent agreement with QM and experiments at a fraction of the cost. GA was developed by Benson in the 1950s to rapidly predict the thermodynamics of organic compounds[28,29]. No such analogous method exists for pure aqueous inorganic compounds. We demonstrate the methodology on group 13 metal clusters which have drawn considerable attention recently due to their importance in catalytic microporous solids[30], green aqueous precursors for large-area metal oxide thin films[2], water treatment[18] as well as tumour diagnosis and possible treatment of malignancies[31]. Second, we use the resulting cluster free energies in the Pourbaix formalism of the Materials Project[32] developed by us[33]. The most important aspect of this formalism is that it uses a combination of computed solid states and clusters with experimentally obtained Gibbs free energies for the monomeric aqueous ions and water, which means that effects such as aqueous ion solvation are inherently accounted for. By combining the GA method with this formalism, we build

a foundation for robustly evaluating the thermodynamic stability of clusters against the competing solid phases and monomeric aqueous ions.

## Results

**Group additivity method for cluster energy prediction.** To demonstrate our approach and study the thermodynamic stability of clusters in aqueous solutions, we consider group 13 metal clusters; in particular, the Al and Ga systems. We emphasize that the monomeric aqueous species are single metal ($M$) aqueous ions, of the form $[MO_yH_z]^{n+}$, whereas the clusters are polymeric metal hydrolysis products $[M_xO_yH_z]^{n+}$. In the case of Al, common monomeric species such as $[Al(H_2O)_6]^{3+}$, $[AlOH(H_2O)_5]^{2+}$ and $[Al(OH)_4]^{4-}$ as well as larger hydrolysis products can be identified by $^{27}$Al-NMR as well as potentiometric studies[34]. Of the multimeric species, the dominant aluminium complex in aqueous systems is the large tridecamer $[Al_{13}O_4(OH)_{24}(H_2O)_{12}]^{7+}$, that is, ε-Keggin ion, found across a broad pH range depending on aluminium ion concentration.

The aqueous behaviour of Ga has significant similarities to Al. Like Al, Ga forms a few monomeric species, such as $[Ga(H_2O)_6]^{3+}$, $[GaOH(H_2O)_5]^{2+}$ and $[Ga(OH)_4]^{4-}$ under different concentration and pH conditions. Ga also forms larger hydrolysis products[14], primarily in the form of the $Ga_{13}$ Keggin cluster, though other hydrolysis products have also been synthesized[35].

To efficiently obtain the energy of a generic cluster, we present a GA method that predicts the thermodynamic stabilities of aqueous group 13 metal hydroxide and oxide clusters, regardless of size or structure. The method quantitatively reproduces first-principles computations in predicting hydrolysis reaction energies and stabilities of both homometallic and heterometallic group 13 clusters with mean absolute error of $3.0\,kcal\,mol^{-1}$. These calculations take seconds, a fraction of the cost of computing the QM values. In this method, the Gibbs free energy of any given aqueous metal oxo/hydroxo cluster is expressed as the sum of the Gibbs free energies of the metal cations and the ligated oxo, hydroxo and aquo ligands. For instance, the absolute Gibbs free energy of Al dimer, $[Al_2(OH)_2(H_2O)_8]$ is:

$$G_{Al_2} = 2G_{Al^{3+}} + 2G_{\mu_2 - OH} + 8G_{\eta - H_2O}. \quad (1)$$

The general hydrolysis reaction can be written as the following equation, where {$M$, $N$} identify the group 13 metals and {$m$, $n$} denote the respective stoichiometries:

$$mM(H_2O)_6^{3+} + nN(H_2O)_6^{3+} \rightarrow M_mN_nO_x(OH)_y(H_2O)_z^{(3m+3n-2x-y)+}$$
$$+ (2x+y)H_3O^+ + (6m+6n-3x-2y-z)H_2O.$$

$$(2)$$

Through inspection of group 13 clusters structures that have either been discovered or conjectured to exist and their QM computed energies[5,36], we identify seven unique ligands that form the basis for replicating any structure (Fig. 1). The Gibbs free energies of the seven basis cluster in Fig. 1 can be written as a set of coupled linear equations with the ligand energies as unknown variables. Using first-principles computed free energies[37–39] for a set of clusters, the Gibbs free energies of the seven common ligands are fit.

**Validation of group additivity method.** The regression plots comparing the GA predicted and QM computed hydrolysis energies are shown in Fig. 2. The linearity between the QM energy and GA predicted energy ($R^2 = 0.99$) shows that the GA method accurately reproduces the QM computed hydrolysis reaction energies of Al, Ga and Al/Ga aqueous oxo/hydroxo clusters. The methodology, the ligand energies and the validation

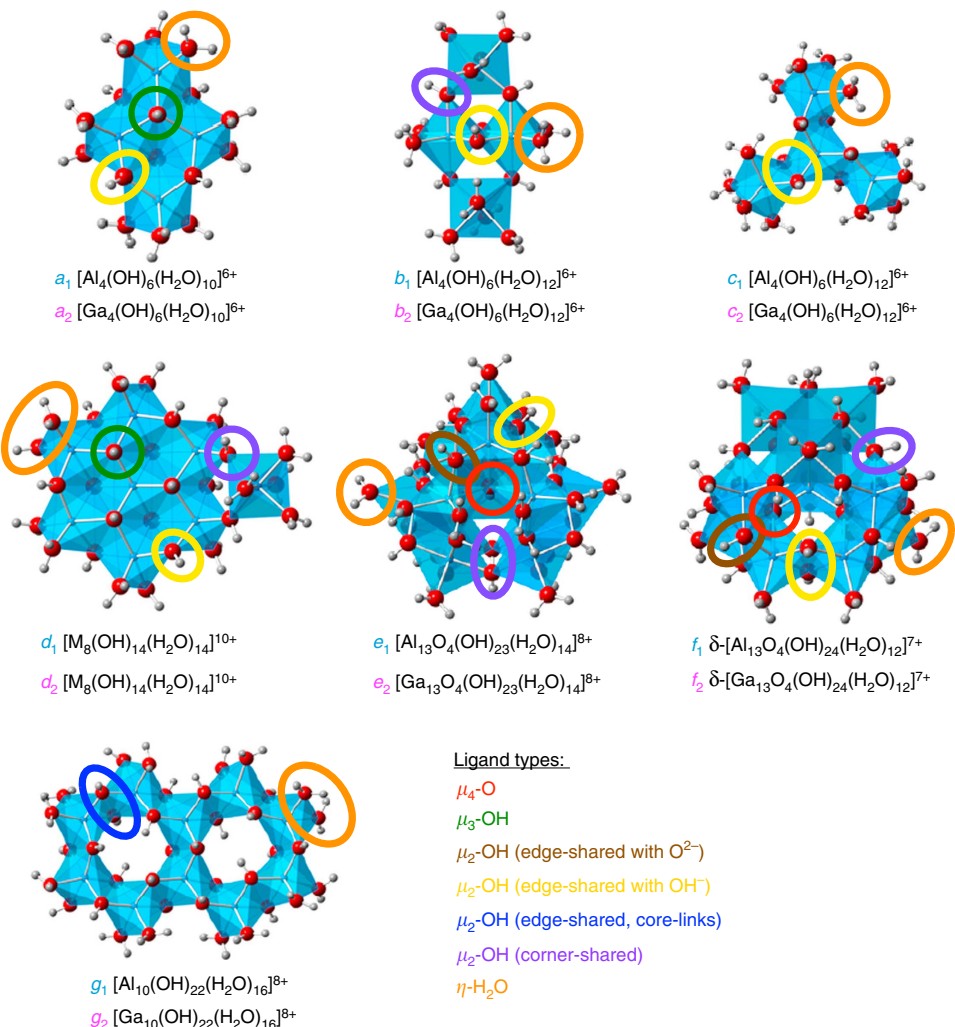

$a_1$ [Al$_4$(OH)$_6$(H$_2$O)$_{10}$]$^{6+}$
$a_2$ [Ga$_4$(OH)$_6$(H$_2$O)$_{10}$]$^{6+}$

$b_1$ [Al$_4$(OH)$_6$(H$_2$O)$_{12}$]$^{6+}$
$b_2$ [Ga$_4$(OH)$_6$(H$_2$O)$_{12}$]$^{6+}$

$c_1$ [Al$_4$(OH)$_6$(H$_2$O)$_{12}$]$^{6+}$
$c_2$ [Ga$_4$(OH)$_6$(H$_2$O)$_{12}$]$^{6+}$

$d_1$ [M$_8$(OH)$_{14}$(H$_2$O)$_{14}$]$^{10+}$
$d_2$ [M$_8$(OH)$_{14}$(H$_2$O)$_{14}$]$^{10+}$

$e_1$ [Al$_{13}$O$_4$(OH)$_{23}$(H$_2$O)$_{14}$]$^{8+}$
$e_2$ [Ga$_{13}$O$_4$(OH)$_{23}$(H$_2$O)$_{14}$]$^{8+}$

$f_1$ δ-[Al$_{13}$O$_4$(OH)$_{24}$(H$_2$O)$_{12}$]$^{7+}$
$f_2$ δ-[Ga$_{13}$O$_4$(OH)$_{24}$(H$_2$O)$_{12}$]$^{7+}$

Ligand types:

$\mu_4$-O
$\mu_3$-OH
$\mu_2$-OH (edge-shared with O$^{2-}$)
$\mu_2$-OH (edge-shared with OH$^-$)
$\mu_2$-OH (edge-shared, core-links)
$\mu_2$-OH (corner-shared)
$\eta$-H$_2$O

$g_1$ [Al$_{10}$(OH)$_{22}$(H$_2$O)$_{16}$]$^{8+}$
$g_2$ [Ga$_{10}$(OH)$_{22}$(H$_2$O)$_{16}$]$^{8+}$

**Figure 1 | The seven aqueous clusters used to compute ligand energies.** Seven metal aqueous oxo/hydroxo clusters were used to compute energies for the seven common ligands. The subscripts of $\mu$ denote the number of metal cations bound to each oxo/hydroxo ligand. Detailed information about the ligands are found in SI.

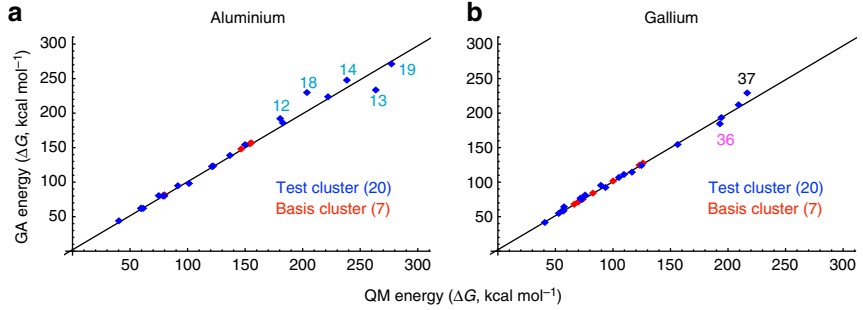

**Figure 2 | GA predicted versus QM computed hydrolysis energies for Al and Ga.** (**a**) Al and (**b**) Ga energies. The slope of the linear regression line is near unity (0.98), the intercept is 2.10 and the $R^2$ value is 0.99. Clusters whose error are larger than 5 kcal mol$^{-1}$ are labelled. Al clusters are in cyan, Ga in magenta and heterometallic clusters in black.

are described in more detail in the Supplementary Tables 1–3, Supplementary Figs 1,2 and Supplementary Notes 1,2.

The GA-Pourbaix formalism is able to quantitatively reproduce not only QM free energies, but also qualitatively the experimental free energies and speciation. To verify the accuracy, the equilibrium constant of the hydrolysis reaction leading to the Keggin aluminium or gallium tridecamer cluster was used to predict speciation. Specifically, the ratio of the monomer and Keggin concentrations was calculated with respect to pH and overall concentration of

metal ions. As shown in Fig. 3, the equilibria predicted from the GA method robustly reproduce the reported experimental speciation behaviour of both γ-Ga and ε-Al Keggin ions.

**Pourbaix diagram for nanoscale clusters.** To introduce the computed clusters to a rigorous thermodynamic framework which allows us to compare their stability with those of competing phases, for example, solids and monomeric aqueous

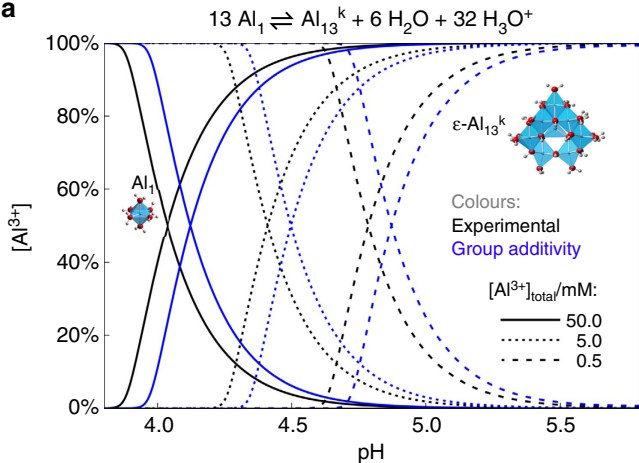

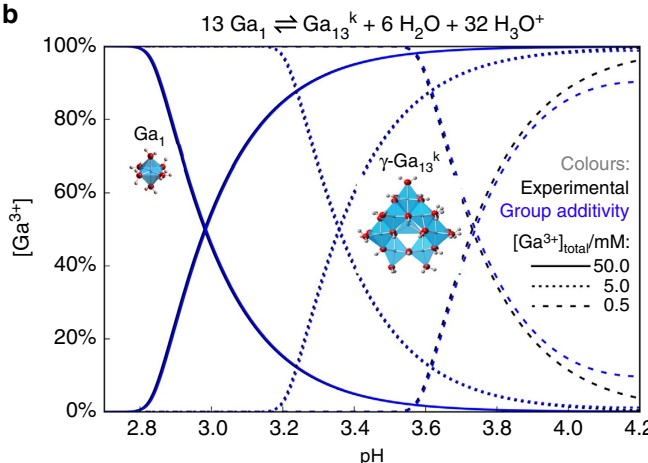

**Figure 3 | Speciation diagrams as a function of pH and concentrations.** Equilibrium constants are deduced from experiments and the group additivity method. (**a**) Aluminum clusters. (**b**) Gallium clusters.

ions, we adopt the Pourbaix formalism, as implemented in the Materials Project[32]. The formation energies for the different species (experimental aqueous ions and first-principles computed solids) are imported through a common reference state scheme allowing for a grand canonical convex hull of stable aqueous and solid species as a function of pH and hydrogen potential. The Pourbaix diagram is constructed by projecting the convex hull to the $E$–pH plane. This formalism has been benchmarked[33] on almost all non-radioactive elements in the periodic table and reproduces or surpasses (due to its access to large amounts of computed solid polymorphs which have not been measured for their thermodynamic properties), the available experimental Pourbaix diagrams. Since launch in 2013, the Materials Project Pourbaix App has been used to successfully predict and interpret the stability of multiple materials and materials classes such as $Mn_2V_2O_7$ (ref. 40), $BiVO_4$ (ref. 41), $MnNiO_3$ (ref. 42), Cu vanadates[41], 2D materials for photocatalyst applications[43] and $CaCO_3$ (ref. 4), as a function of aqueous conditions. In this work we extend the framework by directly importing the cluster energies through equations similar to equation (1). The resulting values for $\mu_{cluster}^{ref}$ are provided in Supplementary Table 4. We note that the Materials Project database contains not only the monomeric aqueous ions and stable solid state phases, but also many more metastable solids—against which the cluster Gibbs free energies are being evaluated, as a function of pH and potential[32]. To date, the Materials Project contains 106 unique

$Al–O_x–H_y$, 41 $Ga–O_x–H_y$ solid compounds and four Al as well as seven Ga monomeric aqueous ions. Furthermore, we emphasize that the aqueous species analyses do not include any counter-ion or salt effects in the aqueous solution.

Figure 4a,b show the two resulting GA-Pourbaix diagrams for aluminium which illustrates the evolution of stable species as a function of pH, standard hydrogen potential—at two representative cluster concentrations. Similar to the well-known experimental aluminium Pourbaix diagram[20,21,44], for pH < 3, the monomer $Al^{3+}$ is stable in solution, while at high pH, the $Al(OH)_4^-$ is preferred. Solid state $Al_2O_3$ is expected to precipitate at neutral conditions and at reducing potentials lower than 2.0 V versus standard hydrogen potential, metallic aluminium represents the equilibrium state in aqueous solutions. Existing experimental Pourbaix diagrams are dominated by solid and monomeric species[20,21]; however, in Fig. 4 we also observe that multimeric aqueous cluster species are predicted to be thermodynamic ground states at slightly acidic conditions. At low cluster concentration ($10^{-8}$ M) we find the ε-Al Keggin (#15 in Supplementary Fig. 2) to be the most stable species between 3.5 < pH < 4.1, while a 'dimer-Keggin' cluster ($Al_{26}^{12+}$ in Fig. 4a) becomes stable at higher pH = 4.1–5.2. This dimer-Keggin resembles a dimer ε-Keggin, but is actually a dimer of δ-Keggins. The δ-Keggin cluster differs from the ε-Keggin by a 60° rotation of one of the four trimeric groups around the exterior of the ε-Keggin cluster. At higher cluster concentrations ($10^{-3}$ M) the dimer-Keggin outcompetes the ε-Keggin, and stabilizes between 3.3 < pH < 4.7. We note that larger oligomeric clusters are favoured at higher cluster concentrations, which suggests a route towards cluster aggregation and subsequent formation of a bulk Al oxide or hydroxide (for example, AlOOH or $Al_2O_3$)[10]. Indeed, the δ-Keggin form is known to polymerize leading to the formation of larger aluminium polycations, whereof $Al_{30}^k$, $Al_{32}^k$ and possibly $Al_{26}^k$, as suggested here, are examples.

Interestingly, for gallium, we find that while the Keggin $Ga_{13}^k$ shows the expected stability regime and speciation with respect to the monomeric $Ga^{3+}$ ions (Fig. 3), in the full Pourbaix diagram it is outcompeted by other species, primarily the solid $Ga_2O_3$ by ~166 meV per Ga at a representative pH = 3.0, and 0.001 M cluster concentration. Hence, at regular concentrations, the gallium Pourbaix diagram—including the clusters—is exactly the same as the traditional one[20,21,44] without any included cluster energies (Fig. 4c). However, by removing the stable phases from the construction of the Pourbaix hull, we simulate scenarios where specific phases are unable to form. For example, if $Ga_2O_3$ nucleation and growth is inhibited, the $Ga_{13}^k$ cluster emerges as the most stable phase for 2.4 < pH < 10.4 (Fig. 4d). At lower pH, the monomeric aqueous ion $Ga^{3+}$ is stable, while for pH >= 10.4, first, the $HGa(OH)_3^{2-}$ and then $GaO_3^{3-}$ form at successively more basic conditions (pH > 11.8).

## Discussion

The two complementary techniques presented in this work provide a robust and efficient framework for calculating the Gibbs free energy and evaluating the thermodynamic stability of clusters in controlled aqueous solutions. For Al, the resulting GA-Pourbaix diagrams advocate the existence of stable nanoscale clusters in aqueous solution which adds pertinent information to the current discussion on non-classical nucleation models[3,7,8]. Figure 4 shows an intuitive succession of the ε-Keggin ion to dimerization of a similar structure (δ-Keggin) as the concentrations of the clusters as well as the monomeric ions are increased. Furthermore, we note that the Al clusters are stabilized without any counter-ion or salt effects, which may be indirectly

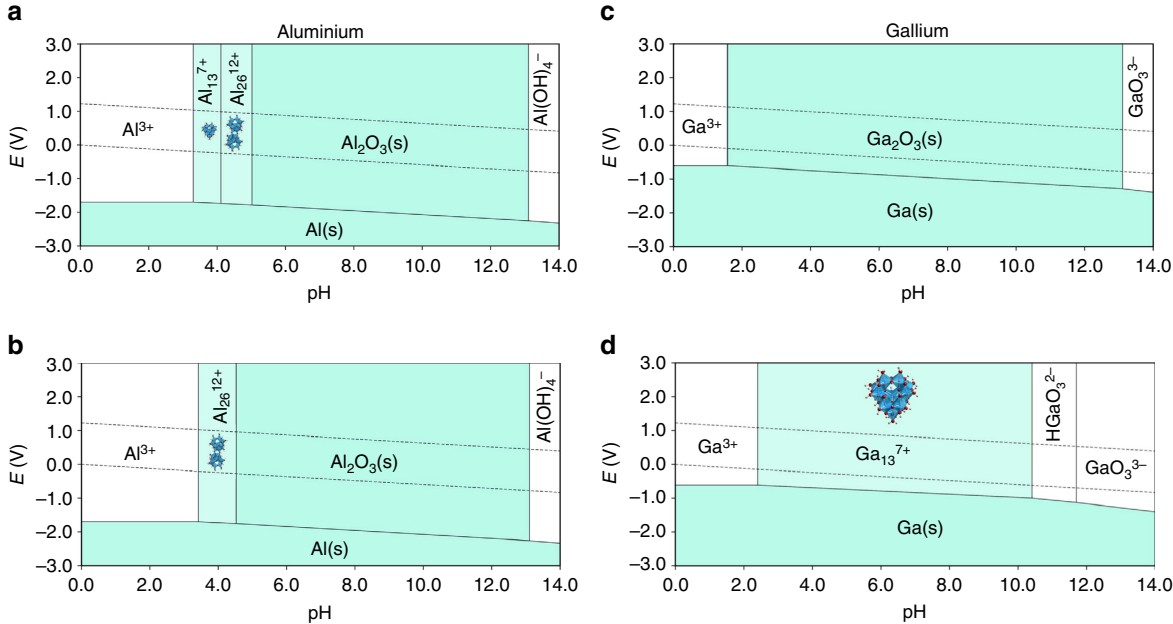

**Figure 4 | Pourbaix diagrams with nanoscale aqueous clusters.** The diagrams incorporate solid phases from first-principles, aqueous ions from experiments and clusters from the GA method. In total, 20 Al clusters are evaluated to produce (**a** and **b**) and 16 Ga clusters are evaluated to to produce (**c** and **d**). In (**a**) the respective concentrations are $10^{-8}$ M for both monomeric and cluster aluminium species and in (**b**) $10^{-2}$ M for monomeric species and $10^{-3}$ M for the aluminium cluster species. In (**c** and **d**), the respective concentrations are $10^{-2}$ M for monomeric species and $10^{-3}$ M for the gallium cluster species. In (**c**) all phases are included whereas in (**d**) we simulate a situation where $Ga_2O_3$ is kinetically inhibited to nucleate which shows the stability regime of the $Ga_{13}$ Keggin. The concentrations for (**a**–**d**) refer to the metal ion and corresponds to the solubility of the stable species at the phase boundary. Hence the concentrations should be treated as a lower limit of solubility at the given potential and pH.

supported by the variety of different solutions in which Al Keggin clusters have been observed[45–48].

For gallium we find that the $Ga_{13}$ Keggin successfully competes with the $Ga^{3+}$ monomeric aqueous ion, as the computed speciation diagram reproduces experimental trends (Fig. 3b). However, in the full GA-Pourbaix diagram, other species such as the aqueous ion $Ga(OH)_2^+$ and the solid $Ga_2O_3$ phase are thermodynamically more stable for the pH range where the Ga clusters may emerge. Hence, the dilute GA-Pourbaix diagram— including cluster energies from 16 Ga clusters—is the same as the traditional experimental Pourbaix diagram[20,21,44]. However, simulating the stability of species under conditions where, for example, we assume $Ga_2O_3$ nucleation to be kinetically hindered, we obtain the GA-Pourbaix diagram in Fig. 4d, where the $Ga_{13}^k$ Keggin stability regime is evident.

The combined results of the aluminium and gallium GA-Pourbaix diagrams prompt us to the following comments. First of all, it is important to note that the results displayed in Fig. 4 assume non-interacting, dilute conditions and do not include any counter-ion effects, which are expected to selectively influence the stability of different clusters. Al Keggin clusters are clearly thermodynamically stable even under these conditions, while Ga clusters may be dependent on salt effects to outcompete the stable phases in the GA-Pourbaix diagram. For example, we stress that the energy difference between the $Ga_{13}^k$ and $Ga_2O_3$ as well as monomeric aqueous specie $Ga(OH)_2^+$ is 166 and 72 meV per Ga; respectively, which is within the energy of counter-ion interactions. In general, analysing the free energy difference between competing clusters as well as other possible species opens up the possibility to rationally influence cluster stability by tuning the solution composition. For example, the 'flat' $Al_{13}$ cluster is predicted to be thermodynamically unstable with respect to the ε-Keggin by 30 meV per Al. However, this cluster has been synthesized and observed under several different conditions[5,49].

In addition, the intermediates that have been suggested to give rise to the 'flat' $Al_{13}$ cluster are not stable from these QM or GA calculations[5]. We speculate that there are many more metastable clusters that may be stabilized by strong salt association and ion-pairing in solution. Available experimental results also suggest that different cluster structures, that may or may not be related to the macroscopic bulk[5,50], form under different aqueous conditions. In particular, nanoscale clusters exhibiting building units resembling metastable bulk materials may pre-condition the synthesis of target nanoparticles[5,11], thin films[2] and bulk materials with select structure and functionality[18].

Finally, we would also like to emphasize that no kinetic effects are included, which are known to be important in many of the observed nucleation processes, and for determining the specific condition-dependent mechanism by which the final crystal and its morphology is formed. Nevertheless, the presented results provide a rigorous thermodynamic framework for the analysis of stable species in solution against which comparison and interpretation of experimental results can be made.

A general predictive computational framework to predict the vast array of aqueous metal clusters found in natural ecosystems, technological applications and biology is a long-standing challenge. In this work, we present the results of combining two procedures to improve our ability to predict and interpret cluster formation in aqueous solutions. First, we disclose a general robust GA method to calculate the free energy of inorganic clusters, which reproduces experimental equilibrium constants and speciation, rival the accuracy of quantum mechanically computed values, and—once developed for a particular chemical system—can be performed in seconds. The speciation diagrams comparing GA, quantum mechanics and experimental results show that the GA method is a general, accurate and expedient method to predict the stabilities of group 13 metal aqua-oxo/ hydroxo clusters. Furthermore, we import the resulting cluster

thermodynamic data into the Materials Project Pourbaix framework, which provides a hybrid approach to classical Pourbaix diagrams—allowing for computed and experimental species to be evaluated within a single-reference state system. The result supports the emerging experimental evidences[3,7–9,12,18] of thermodynamically stable and mildly metastable solvated clusters, which may form as part of pre-nucleation pathways in non-CNT[10]. In the case of Al, we find that the ε-Keggin and 'dimer' δ-Keggin outcompetes monomeric aqueous species at mildly acidic conditions and low cluster concentrations, whereas higher concentrations favour the dimer-Keggin alone. In the case of Ga, we find no cluster species to be more stable than the monomeric ions and the bulk $Ga_2O_3$ solid under dilute, salt-free water conditions. However, the Keggin $Ga_{13}$ cluster is found to be more stable than the monomeric $Ga^{3+}$ under mildly acid conditions and between $2.4 < pH < 10.4$ is the second most stable species, after bulk $Ga_2O_3$. We emphasize that the current framework does not include interactions with other species in solution, such as counter-ions, which are likely to provide an excellent tuning parameter to change the hierarchy of stable clusters and promote nucleation of metastable structures. We also stress that kinetic factors and the actual atomistic path towards nucleation, including structural rearrangement of cluster collectives, are beyond the current analysis. Nevertheless, this GA-Pourbaix methodology will contribute towards improved understanding on nucleation by providing a quantitative framework to aid analysis and ultimately guide rational synthesis of metastable clusters and structures in solution.

## Methods

**Quantum mechanical method for group additivity.** The ground-state structures and Gibbs free energies are computed by HF/6-31G(d,p) with the IEFPCM-UFF continuum solvation model for water. The electronic energy was refined using the B3LYP/6-311 + G(d) single point, and the solvation energy was recomputed using HF/6-311 + G(d) with CPCM-UAKS for water.

**Pourbaix diagram formalism.** To integrate the GA free energies into the Persson Pourbaix formalism we express the clusters formation with respect to the monomeric aqueous ions:

$$mM^{3+}_{(aq)} + nN^{3+}_{(aq)} + (x+y+z)H_2O \rightleftharpoons M_mN_nO_x(OH)_y(H_2O)_z^{(3m+3n-2x-y)+} + (2x+y)H^+.$$ 
$$(3)$$

The corresponding free energy of cluster hydrolysis is then obtained by:

$$\Delta g^{\circ}_{hydrolysis} = \mu^{ref}_{M_mN_nO_x(OH)_y(H_2O)_z^{(3m+3n-2x-y)+}} + 0 - m \cdot \mu^{ref}_{M^{3+}_{(aq)}} - n \cdot \mu^{ref}_{N^{3+}_{(aq)}} - (x+y+z) \cdot \mu^{ref}_{H_2O},$$
$$(4)$$

where $\mu^{ref}_{M_mN_nO_x(OH)_y(H_2O)_z^{(3m+3n-2x-y)+}}$, $0$, $\mu^{ref}_{M^{3+}_{(aq)}}$, $\mu^{ref}_{N^{3+}_{(aq)}}$ and $\mu^{ref}_{H_2O}$ are the chemical potentials of the cluster, protons, metal ions and $H_2O$, respectively. All energies are provided at the standard state, and conform the consistent reference framework by Persson et al.[29] $\mu^{ref}_{M^{3+}_{(aq)}}$, $\mu^{ref}_{N^{3+}_{(aq)}}$ and $\mu^{ref}_{H_2O}$ are known properties, while $\Delta g^{\circ}_{hydrolysis}$ is calculated by the GA method presented here. Hence, $\mu^{ref}_{M_mN_nO_x(OH)_y(H_2O)_z^{(3m+3n-2x-y)+}}$ is obtained from equation (4):

$$\mu^{ref}_{M_mN_nO_x(OH)_y(H_2O)_z^{(3m+3n-2x-y)+}} = \Delta g^{\circ}_{hydrolysis} + m \cdot \mu^{ref}_{M^{3+}_{(aq)}} + n \cdot \mu^{ref}_{N^{3+}_{(aq)}} + (x+y+z) \cdot \mu^{ref}_{H_2O}.$$
$$(5)$$

**Code availability.** The code to plot Pourbaix diagram is available via https://github.com/materialsproject.

**Data availability.** The data that support the findings of this study are available within the paper and its supplementary information file.

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

## Acknowledgements

P.H.-Y.C. is the Bert and Emelyn Christensen Professor of Chemistry at Oregon State University and gratefully acknowledges financial support from the Vicki & Patrick F. Stone Family and the National Science Foundation (NSF, CHE-1352663). L.A.W. acknowledges financial support from OSU Hedberg Fellowship. All authors acknowledge financial support and computing infrastructure in part provided by the NSF Phase-2 CCI, Center for Sustainable Materials Chemistry (NSF CHE-1102637).

## Author contributions

L.A.W. and P.H.-Y.C. developed the group addivity method. L.A.W. performed the quantum mechanical calculations. X.Q. and K.A.P. developed the formalism to integrate Pourbaix diagram with clusters. X.Q. implemented the code base and plotted the diagram. K.A.P., L.A.W., X.Q. and P.H.-Y.C. wrote the final manuscript. All authors participated in the final version of the article.

## Additional information

**Competing interests:** The authors declare no competing financial interests.

