## [Peer Review File · Nature Communications]

Reviewers' comments:

Reviewer #1 (Remarks to the Author):

The consideration of the thermodynamic equilibria of metal oxides with aqueous solution has long suffered from the (previously necessary) assumption that the dissolved ions do not form thermodynamically stable clusters. The growing recognition of the importance of aqueous clusters in a variety of biological and materials systems has made revising these models of equilibria a high priority. In this manuscript, Wills and coworkers present a theoretical framework for doing just that, using Group 13 metal oxide systems as examples. They first illustrate how the computationally calculated hydrolysis energies of various Al/Ga-O-H species can be simply reproduced from sums of terms corresponding to different atomic groups in the clusters. The resulting group additivity relationship allows for a wide range of clusters to be evaluated without the need for an unmanageable number of ab initio calculations. Next, the authors incorporate these group additivity equations into their previously developed formalism for merging theoretical and experimental data in the evaluation of the equilibria between solids and aqueous solution as functions of pH and electrochemical potential to produce Pourbaix diagrams. Finally, this framework is applied to gallium and aluminum oxides systems. In the latter, aqueous clusters emerge as stable for a range of conditions. This paper very beautifully illustrates how theory can be used to elucidate materials growth and dissolution in aqueous solution, and should be read with interest by range of materials scientists and theorists. I thus recommend its publication in Nature Communications once the following issues are addressed:

1. In the abstract, the abbreviation "GA" is used without being previously defined.
2. It was not clear from the text how many distinct cluster configurations were considered in the calculation of the Pourbaix diagrams. The usefulness of the group additivity relationship only becomes apparent when many more cluster geometries are considered than would be feasible with ab initio calculations.
3. I am concerned about how well the group additivity equations will work when counter ions and cluster-cluster interactions are included. The arrangement of groups on the surface of a cluster would likely be very important in terms of how that cluster interfaces with its surroundings. For these reasons, it might be better to qualify the statements about the "generality" of the approach taken here.

Reviewer #3 (Remarks to the Author):

The authors have a key result to report, the development of group-additivity (GA) methods for predicting the stability of aqueous multimeric complexes. The method uncovers potential experimental targets. I would recommend publication in Nature Communications for these developments if well described.

They unfortunately have wrapped this result up into two subthemes of questionable importance and veracity. They need to eliminate these completely as distractions and hype.

1) The claim that this GA method somehow sheds light on pre-nucleation pathways for mineral growth is specious and unnecessary. The authors wave this claim around but provide no link to any known system. It is hype.

a) Liquid-like prenucleation is a big deal in non-classical mineral growth but the structure of the clusters is poorly known, perhaps fleeting (see work by Julian Gale) and the words 'liquid-like' seem to speak loudly.

There is no evidence that any of the clusters identified by the GA methods of this paper have anything to do with a prenucleation cluster, although the fact that they can hydrolyze into amorphous films has been known for decades. The phenomenon also derives from a related phenomenon called 'oriented aggregation' (OA) of Lee Penn and Jill Banfield but again there is no evidence that these clusters have anything to do with OA at all.

b) The authors speak of Pourbaix diagrams as though the addition of metastability is a novel development--even in the Abstract they talk about the diagrams exclusively being constructed of monomeric species, as though they are advancing the field in a fundamental way. Nonsense.

2) Such diagrams including metastable species have been generated for three generations of scientists, beginning with the influential book 'Solutions, Minerals and Equilibria' by Garrels and Christ (1962) immediately after Pourbaix's book. A quick Google search shows this to be true. Look at the bismuth entry in Brookins: Eh-pH Diagrams for Geochemistry confirms this point--it begins with the hexamer.

a) Although the Materials Scientists seem to have recently built a utility for generating diagrams from thermodynamic data (including metastable oligomers) in their Materials Genome Project, the geochemists have done so for many, many decades. The Geochemists Workbench is one such product and there are many others.

Metastability is not new, nor is its inclusion in a phase diagram. Geochemists have built models for solution equilibria up to 5 kbar and 1000 C--Everett Schock has worked hard to include the C-H-O species into the models and these are ultimately unstable within the stability field of water. A quick Google search shows such diagrams with metastable species in the C-H-O, C-H-O-S systems, as well as metals like the bismuth mentioned above, or vanadium.

In constructing a Pourbaix diagram, one chooses the species to include. Usually the metastable equilibria are for solids, but this is only because of the interest of mineralogists. One could just as easily include metastable aqueous clusters such as the decavanadate ion or the Al_{13} cation.

Finally---the summary.

These authors have a great result and one that fully justifies publication in Nature Communications---it is the Group Additivity method and results shown in the Figures.

To write an acceptable paper, the authors need to:

1) Eliminate completely the hype about prenucleation clusters. It is distracting from what is already a fine result.

2) Stop talking about inclusion of multimeric species into a phase diagram as a new development.

These authors have no redox chemistry, so their Figure 4 provides much, much less information about aluminum speciation than Figure 6.4 in Baes and Mesmer, 'Hydrolysis of Cations', which was written in 1976.

Rewrite and resubmit. Don't accept.

Reviewer #4 (Remarks to the Author):

This is an interesting paper, overall, and I believe the novelty of the approach and the generality

of the idea merit publication. Benchmarking against available speciation data supports the validity of the modeling approach and illustrates in an impressive way its success. However, I do have some concerns/queries prior to acceptance.

1) The free energy contributions for group additivity derive from multilinear regression. As such, the individual values will have standard errors associated with them. It would be helpful to report these as this will serve to establish which groups are sufficiently represented to have reliable values and which, if any, may be less certain.

2) I confess to being puzzled by the caption to the Pourbaix diagram figure, which seems to imply something about the enforced molarity of cluster species under consideration. Are not these molarities dictated by the solubility products of the relevant solids together with the pH and applied potential? One cannot "buffer" the monomeric and dimeric species, for example -- the point of the diagram is to illustrate the conditions under which equilibrium favors one over the other. I may certainly be missing something, but it seems as though more explanation is required here.

3) The supporting information makes reference to "the standard state", but offers no details about what that standard state is.

Reviewers' comments:

Reviewer #1 (Remarks to the Author):

The consideration of the thermodynamic equilibria of metal oxides with aqueous solution has long suffered from the (previously necessary) assumption that the dissolved ions do not form thermodynamically stable clusters. The growing recognition of the importance of aqueous clusters in a variety of biological and materials systems has made revising these models of equilibria a high priority. In this manuscript, Wills and coworkers present a theoretical framework for doing just that, using Group 13 metal oxide systems as examples. They first illustrate how the computationally calculated hydrolysis energies of various Al/Ga-O-H species can be simply reproduced from sums of terms corresponding to different atomic groups in the clusters. The resulting group additivity relationship allows for a wide range of clusters to be evaluated without the need for an unmanageable number of ab initio calculations. Next, the authors incorporate these group additivity equations into their previously developed formalism for merging theoretical and experimental data in the evaluation of the equilibria between solids and aqueous solution as functions of pH and electrochemical potential to produce Pourbaix diagrams. Finally, this framework is applied to gallium and aluminum oxides systems. In the latter, aqueous clusters emerge as stable for a range of conditions. This paper very beautifully illustrates how theory can be used to elucidate materials growth and dissolution in aqueous solution, and should be read with interest by range of materials scientists and theorists.

We sincerely thank the referee for his/her careful review, positive statements and for the suggestions which has enabled us to improve the manuscript. We hope that our explanations and modification satisfy the comments/suggestions and summarize the changes made below.

Comment 1: In the abstract, the abbreviation "GA" is used without being previously defined.

Reply: This mistake has been corrected. "GA" in the abstract is now "Group Additivity".

Comment 2: It was not clear from the text how many distinct cluster configurations were considered in the calculation of the Pourbaix diagrams. The usefulness of the group additivity relationship only becomes apparent when many more cluster geometries are considered than would be feasible with ab initio calculations.

Reply: It is an excellent point that the number of clusters for Pourbaix diagrams is an important information and be clear to the readers. We have added the number of clusters to captions in figure 4. We completely agree that group additivity relationship would be more indispensable when there is intractable number of cluster. We would like to respectfully clarify that 1) our group additivity relationship in this work is develop as a method applicable to the broader community; 2) while the number clusters evaluated in this work is feasible with ab initio calculation on state of art supercomputers, we are expecting more clusters to be studied in the future works. The following text is added to figure 4 caption:

“In total, 20 Al clusters are evaluated to produce (a and b) and 16 Ga clusters are evaluated to produce (c) and (d).”

Comment 3: I was concerned about how well the group additivity equations will work when counter ions and cluster-cluster interactions are included. The arrangement of groups on the surface of a cluster would likely be very important in terms of how that cluster interfaces with its surroundings. For these reasons, it might be better to qualify the statements about the "generality" of the approach taken here.

Reply: We agree that there may be important effects from the counter ions and cluster-cluster interactions. Future work will focus on the effects of these two factors, but for now we have included some discussion of these effects at the end to make clear the limits of GA.

Reviewer #3 (Remarks to the Author):

The authors have a key result to report, the development of group-additivity (GA) methods for predicting the stability of aqueous multimeric complexes. The method uncovers potential experimental targets. I would recommend publication in Nature Communications for these developments if well described. (...)

We are very grateful for the reviewer's appreciation of our paper and for the instructive comments. We offer the following clarifications and comments and subsequent modifications of our manuscript.

Comment 1: The authors speak of Pourbaix diagrams as though the addition of metastability is a novel development--even in the Abstract they talk about the diagrams exclusively being constructed of monomeric species, as though they are advancing the field in a fundamental way. Nonsense.

Such diagrams including metastable species have been generated for three generations of scientists, beginning with the influential book 'Solutions, Minerals and Equilibria' by Garrels and Christ (1962) immediately after Pourbaix's book. A quick Google search shows this to be true. Look at the bismuth entry in Brookins: Eh-pH Diagrams for Geochemistry confirms this point--it begins with the hexamer.

Reply: We definitely do regret any indication of being 'first' to include multimeric cluster species and/or metastable species/compounds in the construction of Pourbaix diagrams. Furthermore, we offer the following comments on the subject of thermodynamically stable multimeric aqueous cluster species:

We are thankful to the reviewer for pointing out *Eh-Ph Diagram for Geochemistry, Brookins* (1987). Indeed, there is a $\text{Bi}_6\text{O}_6^{6+}_{(\text{aq})}$ cluster found to be stable in the Bi Pourbaix diagram at acid, oxidative conditions, however, we believe that it is the *only* element (spanning 61 elements in the periodic table) in *Brookins* that exhibits a stable multimeric specie beyond dimers in the published Pourbaix diagrams. Conspicuously, known cluster forming systems such as Hf, Al,

Ga, Nb are all shown as monomeric aqueous Pourbaix diagrams and there is no thermochemical data for any multimeric aqueous species included for these systems in *Brookins*. Thermochemical data for multimeric species are available for the Bi, V and Cr systems, but indeed only $\text{Bi}_6\text{O}_6^{6+}(\text{aq})$ shows a resulting stability region in the Pourbaix diagram.

Conversely, in *Atlas of Eh-pH diagrams, Intercomparison of thermodynamic databases by National Institute of Advanced Industrial Science and Technology, Takeno 2005* ; none of the Bi Pourbaix diagrams based on the FactSage, SUPCRT, LLNL and JNC-TBD databases exhibit any stability region of a Bi aqueous ion cluster specie in the resulting Pourbaix diagrams. This despite the fact that several multimeric aqueous Bi clusters – including the $\text{Bi}_6\text{O}_6^{6+}(\text{aq})$ - are included in FactSage.

Indeed, in *Atlas 2005*, while some of the cited databases contain thermochemical data on diverse aqueous ion clusters up to 13 metal cations, none of the 189 Pourbaix diagrams spanning the 86 covered elements in the periodic table exhibit any stability regime for any aqueous ion cluster specie with more than 3 cations (at 10^{-10} M). The only two that show a stable multimeric aqueous ion specie is the FactSage iodine Pourbaix diagram, exhibiting a stable $\text{I}_2\text{OH}^-(\text{aq})$ dimer at oxidizing conditions, $2 < \text{pH} < 8$ and the LLNL uranium Pourbaix diagram, exhibiting a stable trimer $(\text{UO}_2)_3(\text{OH})_7^-$ at oxidizing, alkaline conditions. We speculate that one reason for the difference between the experimental diagrams (e.g. Bi in Brookins 1987 vs Bi in FactSage 2005) reflects the sensitivity to very small energy differences and the difficulty in experimentally obtaining those free energy differences accurately for aqueous clusters, removing counter-ion and other ligand effects. In this aspect, computations may offer a complementary approach.

Hence, while we are completely in agreement that multimeric species have been included in the construction of Pourbaix diagrams before we are unable to find compelling evidence that thermodynamically stable multimeric aqueous ion clusters – as evidenced by Pourbaix diagrams - are well established in the literature. In response to the comment, in the resubmitted manuscript we have:

- i) Rephrased sentences to remove any indication of being 'first' to include multimeric ions in the construction of Pourbaix diagrams
- ii) Made sure to point out the existence of Pourbaix diagrams, citing *Brookins* and *Atlas*, including multimeric aqueous cluster species in the construction of Pourbaix diagrams.

Comment 2: The claim that this GA method somehow sheds light on pre-nucleation pathways for mineral growth is specious and unnecessary. The authors wave this claim around but provide no link to any known system. (...) Liquid-like prenucleation is a big deal in non-classical mineral growth but the structure of the clusters is poorly known, perhaps fleeting (see work by Julian Gale) and the words 'liquid-like' seem to speak loudly. There is no evidence that any of the clusters identified by the GA methods of this paper have anything to do with a prenucleation cluster, although the fact that they can hydrolyze into amorphous films has been known for decades. The phenomenon also derives from a related phenomenon called 'oriented aggregation' (OA) of Lee Penn and Jill Banfield but again there is no evidence that these clusters have anything to do with OA at all.

Reply: We do recognize that the term ‘pre-nucleation cluster’ is used somewhat differently in different communities and the perceived role of these clusters in the nucleation process is an active topic of research. To exemplify we provide a few excerpts below:

In the three first examples below, Keggin-like and well-structured molecular nanoclusters in solution are described as pre-nucleation species or precursors to nucleation.

[Nyman et al, Science 2015 347, 6228, 1359]

“The identified non-classical growth behavior of iron oxides in both nature (3) and the laboratory (12)—defined by aggregation of pre-nucleation clusters rather than atom-by-atom growth—supports the existence of a discrete Fe₁₃ ion as a precursor to ferrihydrite and magnetite”

[Navrotsky et al PNAS 2011, 108 (36), 14775–14779]

“Underscoring the importance of ϵ -Al₁₃ clusters in natural and anthropogenic systems, these data provide conclusive thermodynamic evidence that the Al₁₃ Keggin cluster is a crucial intermediate species in the formation pathway from aqueous aluminum monomers to aluminum hydroxide precipitates.”

Similarly, in [Nonclassical nucleation and growth of inorganic nanoparticles by Hyeon et al Nature Reviews Materials 1, Article number: 16034 (2016)] the terms pre-nucleation cluster, magic cluster, nanocluster and molecular cluster are used alternately describing the range of cluster species formed during non-classical nucleation.

On the other hand, Colfen and Gebauer make a distinction between the more dynamic (liquid-like) pre-nucleation clusters and more defined Keggin-like clusters:

[Colfen et al, : Chem. Soc. Rev., 2014, 43, 2348]

“From the point of view of dynamics (iv), the polynuclear Al₁₃⁺ Keggin ion may hence not be regarded strictly a solute, thereby not qualifying as a PNC (sic pre-nucleation cluster). In this sense, it may be rather regarded as a nanosolid that has formed from smaller oligomeric PNC precursors. Consistently, the Keggin ion will undergo aggregation as a result of the creation of energetically unfavourable interfacial surfaces during aluminium hydroxide precipitation.”

Colfen and Gebauer continues to propose to denote Keggin and similar structured aqueous ions as nanoscopic droplets, which aggregate and form larger entities in the sequence of non-classical nucleation:

“For example, the crystallisation of calcium carbonate along the PNC pathway can be understood chemically and structurally as a progressive, step-wise loss of hydration water, according to the sequence Ca²⁺(aq)/HCO₃⁻(aq)/CO₃²⁻(aq) - PNCs - dense liquid nanodroplets - liquid ACC - solid ACC - anhydrous crystalline polymorphs.”

We certainly do not want to over-state our results, and agree that the term ‘pre-nucleation’ may have been used too liberally in our paper. Hence, we have replaced the term ‘pre-nucleation’ with ‘nanoscale’ cluster as a description of the clusters studied, and amended the sentences which discusses the possible connection between thermodynamic stability of clusters with non-classical nucleation, as supported by the cited work above. For example:

For Al, the resulting GA-Pourbaix diagrams ~~strongly support the recently proposed non-classical nucleation models~~^{3,7,8} ~~advocating~~ the existence of stable nanoscale ~~pre-nucleation~~ clusters in aqueous solution which adds pertinent information to the current discussion on non-classical nucleation models.^{3,7,8}

In conclusion, we sincerely thank the reviewer for helping us to improve upon our manuscript, and hope to have clarified any disparities.

Reviewer: 4; Recommendation

Comment 1: The free energy contributions for group additivity derive from multilinear regression. As such, the individual values will have standard errors associated with them. It would be helpful to report these as this will serve to establish which groups are sufficiently represented to have reliable values and which, if any, may be less certain.

Reply: The values for the free energy contributions for group additivity do not derive from multilinear regression. Instead, these values were derived from systems of linear equations where the variables in the system of equations represent the ligand types and metal types. We actually used two systems of linear equations with 8 variables each; one system represented the ligand and metal free energy contributions for aluminum clusters and the other represented the ligand and metal free energy contributions for gallium clusters. As such, these free energy contributions do not have a standard error associated with them. In order to allow the reader a better understanding which ligands have more or less reliable values, however, we have included the raw values in the new version of the SI. The degree to which the same ligand energies differ between Ga and Al clusters should provide a qualitative picture into which ligands have more precise energies than others.

Comment 2: “I confess to being puzzled by the caption to the Pourbaix diagram figure, which seems to imply something about the enforced molarity of cluster species under consideration. Are not these molarities dictated by the solubility products of the relevant solids together with the pH and applied potential? One cannot "buffer" the monomeric and dimeric species, for example -- the point of the diagram is to illustrate the conditions under which equilibrium favors one over the other. I may certainly be missing something, but it seems as though more explanation is required here.”

Reply: Thanks for pointing out the confusing caption. We gratefully take the opportunity to add more explanation. We agree that solubility at a given point (pH, E) determines maximum molarity. However, the purpose of the current Pourbaix diagram procedure is to map out stable range for each species, rather than finding the highest possible concentration at a given (pH, E).

The concentration settings in figure 4 corresponds to the solubility of the stable species phase boundary, and should be treated as a lower limit of the solubility in that area. The following text is added to figure 4 caption to clarify it.

“The concentrations for (a-d) refer to the metal ion and corresponds to the solubility of the stable species at the phase boundary. Hence the concentrations should be treated as a lower limit of solubility at the given potential and pH.”

Comment 3: “The supporting information makes reference to "the standard state", but offers no details about what that standard state is.”

Reply: We thank the reviewer for the good suggestion. The following details about standard state has been added to supporting information table S2:

“(room temperature 25 °C, atmospheric pressure, and 1 M concentration)”

Reviewers' comments:

Reviewer #1 (Remarks to the Author):

The authors have addressed my concerns in their revised manuscript.

Reviewer #3 (Remarks to the Author):

The authors have a fine contribution to make to the aqueous oxide literature about estimating cluster thermodynamics using Group Additivity.

I am a big fan of this paper and want to see it published in Nature Communications.

However, the revised paper, and the responses to the referees, indicate that the authors STILL overestimate the significance of their result for Pourbaix diagrams.

They don't know the literature and it shows.

1) In receiving their revised manuscript ten minutes ago I just flipped open to page 337 of 'Aqueous Chemistry of the Elements' by Schweitzer and Pesterfield (2010) to see full treatment of the multimeric vanadate species at total vanadium concentration of 0.1 M, with all of the decavanadates. I am not even trying to conduct an exhaustive search.

2) Because their system is Group 13 metals, there is no redox chemistry within the stability field of water. Thus their Figure 4 shows much, much less information than Figure 6.4 of Baes and Mesmer (1976) (p. 122), which shows a rich array of multimeric hydrolysis complexes.

Please clean this up. Claims like these detract from what is otherwise a fine paper.

I also suggest you have a look at the following free software which can be used for calculating Pourbaix diagrams and includes many sets of aqueous multimers:
<https://www.kth.se/en/che/medusa/downloads-1.386254>

Reviewer #4 (Remarks to the Author):

I am satisfied with the revisions that have been made and can recommend publication.

Reviewers' comments:

Reviewer #1 (Remarks to the Author):

The authors have addressed my concerns in their revised manuscript.

Response: We sincerely thank the referee for his/her positive statements and for the suggestions which has enabled us to improve the manuscript.

Reviewer #3 (Remarks to the Author):

The authors have a fine contribution to make to the aqueous oxide literature about estimating cluster thermodynamics using Group Additivity. I am a big fan of this paper and want to see it published in Nature Communications. However, the revised paper, and the responses to the referees, indicate that the authors STILL overestimate the significance of their result for Pourbaix diagrams. They don't know the literature and it shows.

Comment 1: In receiving their revised manuscript ten minutes ago I just flipped open to page 337 of 'Aqueous Chemistry of the Elements' by Schweitzer and Pesterfield (2010) to see full treatment of the multimeric vanadate species at total vanadium concentration of 0.1 M, with all of the decavanadates. I am not even trying to conduct an exhaustive search.

Response: We appreciate the reviewer's instructive comments. Going through 'Aqueous Chemistry' which spans the periodic table, we indeed identify 3 elements which are shown to exhibit stable multimeric species in the Pourbaix diagram; Mo, V and W (up to 7 metal ions). Again, we note that Al, Ga, Nb, Hf, Sn, Bi, etc and other systems which are likely cluster forming are presented with only monomeric aqueous species (and no multimeric data is included in the Pourbaix analysis). However, we are delighted to include more experimental evidence, e.g. that multimeric species are found to be stable in W, Mo and V at specific concentrations, and have included sentences to that effect:

While the majority of published Pourbaix diagrams exhibit exclusively monomeric aqueous ion stability domains, growing thermochemical data on clusters is being incorporated into speciation and Pourbaix analyses. [Brookins, Atlas and Aqueous Chemistry] As examples, we note that stable multimeric species, beyond dimers and trimers, are suggested for Mo, W and V [Aqueous Chemistry] and for Bi [Brookins] at 0.1 M and high potential. On the other hand, known cluster-forming systems such as Al, Fe, Ga, Hf etc present only monomeric aqueous ion stability in these reference works.

We have carefully re-read our paper, and rephrased language that may be considered too strong. For example, the title (again) has been reworded and the abstract has been amended.

Comment 2: Because their system is Group 13 metals, there is no redox chemistry within the stability field of water. Thus their Figure 4 shows much, much less information than Figure 6.4 of Baes and Mesmer (1976) (p. 122), which shows a rich array of multimeric hydrolysis

complexes. Please clean this up. Claims like these detract from what is otherwise a fine paper. I also suggest you have a look at the following free software which can be used for calculating Pourbaix diagrams and includes many sets of aqueous multimers:

<https://www.kth.se/en/che/medusa/downloads-1.386254>

Response: Indeed, all the thermodynamic information underlying Fig 6.4 in Baes and Mesmer is of course available, as the equilibrium quotients are obtained from the free energies of the ions, the solids and water. For the formalism to combine computed species with experimental results, see Persson et al, *Prediction of solid-aqueous equilibria: Scheme to combine first-principles calculations of solids with experimental aqueous states*, Phys. Rev. B 85, 235438 (Ref 33). Thus, figures similar to 6.4 in Baes and Mesmer can easily be plotted if desired. Our paper is intended to show the value of our methodology which combines rapid evaluation of *any* cluster energy within a certain chemistry, a priori and without counter-ions, with the Pourbaix formalism which incorporates experimental data for known aqueous ions and the 67,000 calculated solid compounds available in the Materials Project. We believe this capability significantly extends our knowledge of multi-component Pourbaix diagrams, and even more so as it now can include clusters – both those known from experiments and new ones from computations.

We sincerely encourage the reviewer to visit <https://www.materialsproject.org/#apps/pourbaixdiagram>, where diagrams up to three elements can be explored online with all the aqueous ions from FactSage, as a function of concentration, pH and potential. We are excited to include more aqueous ions, and aqueous clusters as a result of our paper and the interaction with the reviewer, to highlight the rich cluster chemistry and its important role in aqueous environments.

In conclusion, we thank the reviewer for helping us to improve upon our manuscript, and hope to have arrived at a mutually acceptable description of our work.

Reviewer #4 (Remarks to the Author):

I am satisfied with the revisions that have been made and can recommend publication.

Response: We sincerely thank the referee for his/her positive statements and for the suggestions which has enabled us to improve the manuscript.

REVIEWERS' COMMENTS:

Reviewer #3 (Remarks to the Author):

I am satisfied with the changes and urge acceptance. The Group Additivity work is a real advance.

Response to Reviewers - Manuscript Version 3

Response to Reviewer #3:

We sincerely thank the referee for his/her positive statements and for the suggestions which has enabled us to improve the manuscript.